# Subspace-Based Two-Step Iterative Shrinkage/Thresholding Algorithm for Microwave Tomography Breast Imaging

**DOI:** 10.3390/s25051429

**Published:** 2025-02-26

**Authors:** Ji Wu, Fan Yang, Jinchuan Zheng, Hung T. Nguyen, Rifai Chai

**Affiliations:** 1School of Engineering, Swinburne University of Technology, Hawthorn, VIC 3122, Australia; jwu1@swin.edu.au (J.W.); jzheng@swin.edu.au (J.Z.); hungnguyen@swin.edu.au (H.T.N.); 2Sichuan Canyearn Medical Equipment Co., Ltd., Chengdu 610000, China; yangfan@wpeony.com; 3Shenzhen Peini Digital Technology Co., Ltd., Shenzhen 518000, China

**Keywords:** microwave tomography, dielectric properties, breast health, distorted born iterative method

## Abstract

Microwave tomography serves as a promising non-invasive technique for breast imaging, yet accurate reconstruction in noisy environments remains challenging. We propose an adaptive subspace-based two-step iterative shrinkage/thresholding (S-TwIST) algorithm that enhances reconstruction accuracy through two key innovations: a singular value decomposition (SVD) approach for extracting deterministic contrast sources, and an adaptive strategy for optimal singular value selection. Unlike conventional DBIM methods that rely solely on secondary incident fields, S-TwIST incorporates deterministic induced currents to achieve more accurate total field approximation. The algorithm’s performance is validated using both synthetic “Austria” profiles and 45 digital breast phantoms derived from the UWCEM repository. The results demonstrate robust reconstruction capabilities across varying noise levels (0–20 dB SNR), achieving average relative errors of 0.4847% in breast tissue reconstruction without requiring prior noise level knowledge. The algorithm successfully recovers complex tissue structures and density distributions, showing potential for clinical breast imaging applications.

## 1. Introduction

Breast cancer stands as the prevailing form of cancer among women globally, imperiling countless lives annually. The World Health Organization reported 2.3 million fresh cases of the ailment and 685,000 fatalities associated with breast cancer in 2020, highlighting significant disparities across nations and regions [1]. In Australia, breast cancer ranks as the second most common type of cancer and the fifth most frequent reason for cancer-induced fatalities during 2020, with an approximated 20,640 novel cases and 3214 casualties in 2022 [2]. The expedited discovery and precise diagnosis of breast cancer prove decisive in bolstering the subsistence and living quality of patients.

Imaging techniques for the breast are used either for routine screening or precise diagnosis. Common forms of breast imaging include x-ray mammography and magnetic resonance imaging (MRI), both of which depend either on ionizing radiation or robust magnetic fields to capture detailed pictures of breast structure. Ultrasound, which stands apart as it does not use radiation, instead employs high-frequency sound waves to visualize breast tissues. Yet, these traditional approaches are not without flaws: inconsistencies in sensitivity and specificity, costly procedures, invasiveness, reliance on the operator’s skill, and potential health hazards are among the chief concerns. Therefore, there is a growing demand for alternative or supplementary imaging methods that can sidestep these issues and shed more light on the characteristics of breast tissues.

Microwave imaging is a promising imaging technique [3] that employs non-ionizing electromagnetic waves within the microwave frequency range from 500 MHz [4] to 9 GHz [5] to examine the dielectric properties of breast tissue, indicating its physiological and pathological conditions. Compared to traditional approaches, microwave imaging has several advantages, including non-radiative, non-invasive, and portability [6,7]. Microwave imaging may be categorized into two types: microwave tomography, which focuses on reconstructing the complex permittivity distribution of breast tissue via solving an inverse scattering problem (ISP), and radar-based microwave imaging, which utilizes the reflected wave from objects to reconstruct an image. Radar-based imaging system leverages the differential electrical dielectric properties between normal and malignant breast tissues to generate reflections when transmitting microwaves through the internal breast with a tumor. In recent years, numerous algorithms have been devised. A Huygens Principle-based method is employed to process the microwave signals and to build the respective microwave images [8]. This prototype was tested on phantoms and then clinically on 51 breasts by using a metric that calculates the ratio between the image intensity’s maximum and average values. To improve the performance of algorithms and obtain desirable image quality, several artifact removal techniques were presented and verified on either digital breast or physical breast phantoms. Elahi [9] proposed an entropy-based adaptive method to group signals with similar artifacts and then remove the artifact from each group separately using a hybrid artifact removal algorithm. In 2024 [10], a new pre-artifact removal algorithm is suggested for microwave imaging that can identify multiple tumors, eliminate pre-artifact, and improve image quality. The algorithm comprises an adaptive filter and threshold that allow for precise identification of both tumors and surface interference. The utility of radar-based microwave imaging is quite extensive as it includes a range of applications, such as investigating cerebral changes resulting from Alzheimer’s disease [11], identifying hemorrhagic strokes [12], and assessing the progression of whole-brain atrophy [7].

Different from radar-based microwave imaging, microwave tomography aims to reconstruct a profile of the breast by solving the ISP of its dielectric characteristics. To obtain a reconstructed image of the unknown object’s position, shape, and dielectric parameters in the domain of interest (DOI), microwave tomography employs inverse scattering techniques that capitalize on scattering signals, from various scatters. It is also known as the challenge of microwave tomography that electromagnetic ISP is nonlinear and ill-posed [13]. Therefore, the reconstruction algorithm often includes nonlinear optimization and regularization terms to reconcile the nonlinearity and ill-posedness of the ISP [14]. In recent decades, numerous algorithms have been proposed. In the 1990s, algorithms using conjugate gradient (CG), Levenberg–Marquardt, and Newton-Kantorovich methods [15,16,17] were introduced to solve 2D ISPs. During the same period, two other classic nonlinear algorithms were proposed: the Born iterative method (BIM) [18] and the distorted Born iterative method (DBIM) [19]. In the BIM, the Green’s function remains constant in each iteration. In contrast, in the DBIM, starting from the second iteration, the Green’s function is recalculated based on the dielectric properties reconstructed from the previous iteration. Many derivative algorithms have been developed from DBIM, such as the two-step iterative shrinkage/thresholding method (TwIST) based on DBIM, which has been applied in biomedical imaging [20,21]. Zhang et al. [22] developed a two-step iterative microwave tomography approach that addresses overfitting and artifacts through multi-frequency and multi-resolution processing, achieving millimeter-level accuracy in brain hemorrhage detection. Lu and Kosmas [23] extended this concept to three-dimensional head imaging, implementing GPU-accelerated FDTD with DBIM and demonstrating superior performance compared to 2D implementations in scenarios with limited prior information. For bone health monitoring, Amin et al. [24] employed DBIM with compressed sensing-based linear inversion, successfully differentiating between osteoporotic and osteoarthritis bones even under low SNR conditions. Alongside conventional threshold-based strategies, there has been growing interest in adaptive or multi-threshold iterative shrinkage approaches for microwave breast imaging. In particular, Ambrosanio et al. [25] proposed a multi-threshold iterative DBIM-based algorithm that adaptively adjusts multiple threshold values to better capture the heterogeneous nature of breast tissue. Their work demonstrated improved reconstruction performance in two-dimensional scenarios when compared to single-threshold methods.

The abovementioned algorithms fall into the category of field-type methodologies, which exclude the consideration of equivalent induced currents during the reconstruction process. Starting from 2000, source-type algorithms have been proposed and developed by researchers. Contrast source inversion [26,27,28] is an algorithm that approaches the contrast collectively to determine the induced currents. Meanwhile, the subspace-based optimization method (SOM) [29] employs a mathematical approach to divide the induced currents into two components: those originating from the inhomogeneous background (deterministic part) and those produced by the scattering contrast (unsolved part). In [29], the Green’s function and the scattering field are employed to extract the deterministic component of the induced current. Then, the unresolved portion of the induced current is resolved using the CG method. By incorporating the deterministic induced current, the approximation of the total electric field becomes more precise, resulting in enhanced convergence speed.

It reported that DBIM is sensitive to the initial guess [30]. Recently, we have found that the subspace-based two-step iterative shrinkage/thresholding method (S-TwIST) [31] is less dependent on an initial guess for 2D simple geometrical reconstruction. This algorithm retrieves the deterministic part of the contrast source from the Green’s function and scattered field signals, using the singular value decomposition (SVD). A more accurate approximation can be obtained by summing the secondary incident field and scattered field using the deterministic part of the contrast source; thus, S-TwIST can recover scatterers that are more accurate than the original TwIST.

In this paper, we propose an extension of S-TwIST and introduce a new optimization strategy to enhance it. Initially, the ’Austria’ profile is proposed to verify the reconstruction performance of the S-TwIST algorithm. In the proposed algorithm, the deterministic part of the induced current is retrieved using thin SVD linearly. The singular value matrix is arranged in descending order, and the algorithm can adjust to use the first largest integer *L* of singular value, where the minimum value of *L* is 0, and the maximum value is determined by the number of receivers. In reality, an ideal environment without noise is difficult to achieve. A series of experiments are designed to verify the performance of S-TwIST under different noise environments and different choices of *L* values. The results show that excessively high *L* values can lead to instability in the optimization process when in noisy scenarios. Secondly, choosing an appropriate *L* value makes S-TwIST more robust in noisy environments.

In the experiments mentioned above, the optimal *L* value varies inversely with noise levels. However, real imaging systems encounter random noise levels, making fixed *L* values potentially suboptimal for algorithm performance. To address this challenge, we propose an adaptive strategy for *L* value selection under unknown noise conditions. Rather than using a fixed *L* value, our approach evaluates multiple *L* values within a predefined range during each iteration. The solution corresponding to the minimum cost function is then selected as the optimal result. Tests demonstrate that this adaptive strategy enhances algorithm performance and improves robustness in noisy environments.

Finally, to demonstrate the potential of the adaptive S-TwIST in microwave breast tomography imaging, we reconstructed 2D breast phantoms using the UWCEM breast model repository [32]. This repository provides nine different types of breast phantoms derived from MRI scans. Compared to the reconstruction of synthetic geometric shapes, breast reconstruction presents more complex shapes and details, indicating a greater challenge. The results demonstrate the algorithm’s effectiveness in reconstructing complex breast tissue structures with varying densities, achieving an average relative error of 0.4847% across all samples.

## 2. Methods

### 2.1. S-TwIST Microwave Tomography Algorithm

#### 2.1.1. Forward Solver

For the electromagnetic (EM) scattering problem, we consider a two-dimensional setup, as illustrated in Figure 1. This configuration features a circular sensor array placed around a domain of interest (DOI), denoted by *D*. The sensors function alternately as transmitters and receivers, collecting scattered field measurements for image reconstruction.

The electromagnetic wave interaction with the imaging domain is numerically simulated using the Finite-Difference Time-Domain (FDTD) approach. This computational method implements a spatial and temporal discretization scheme to solve Maxwell’s equations, enabling accurate modeling of wave propagation phenomena [33]. The algorithm progresses through discrete steps, updating both electric and magnetic field components on a structured mesh. The total field distribution throughout the domain is determined by incorporating the material’s dielectric properties and appropriate boundary conditions into the FDTD framework. In this study, we focus on the two-dimensional transverse magnetic polarization along the z-direction (TMz), where the scattered field can be described by an integral equation. The relationship between the total field ***E^t^****^ot^* and the incident field ***E****^inc^* is governed by the following:(1)$$E^{tot}(r,r\prime \prime ;\omega )=E^{inc}(r,r\prime ;\omega )+\int_{D}G(r\prime \prime ,r;\omega )\cdot \chi (r;\omega ){\cdot E}^{tot}(r,r\prime ;\omega ),$$
where the vector **r** ∈ *D* denotes the source point in the domain *D*. The vectors **r**’ ∈ *S* and **r**″ ∈ *S* denote the location of the transmitters and receivers outside the DOI. The contrast function defined as ***χ***(**r**;*ω*) = *ε*_r_(**r**)/*ε*_b_ − 1 − j(*σ*(**r**) − *σ*_b_)/*ωε*_b_. The interaction between source points and observation points in the background medium is described by Green’s function:(2)$$G\left(r\prime \prime ,r;\omega \right)=-\frac{jk_{b}^{2}}{4}H_{0}^{\left(2\right)}\left(k_{b}\left|r-r\prime \prime \right|\right),$$
where *k_b_* is the wavenumber in the background, once the total field is computed, the scattered field at **r**″ outside the domain *D* satisfies the following integral equation:(3)$$E^{sca}(r\prime ,r\prime \prime ;\omega )=\int_{D}G(r\prime \prime ,r;\omega )\cdot \chi (r;\omega )\cdot E^{tot}(r,r\prime ;\omega ).$$

For numerical implementation, we discretize the imaging domain *D* into *N_i_* elements. This discretization allows us to reformulate Equations (1) and (2) into matrix form:(4)$$E^{tot}={\left(I-G_{D}\cdot \chi \right)}^{-1}\cdot E^{inc},$$(5)$$E^{sca}=G_{S}\cdot \chi \cdot {\left(I-G_{D}\cdot \chi \right)}^{-1}\cdot E^{inc},$$
where ***G_D_*** represents the Green’s function operator mapping fields within the domain, and ***G_S_*** describes the field propagation to measurement points, and ***I*** is the identity matrix.

The measurement system configuration employs a circular antenna array consisting of 30 uniformly distributed elements surrounding Domain *D*. In the FDTD implementation, each antenna element is modeled as a line source perpendicular to the imaging plane. When one antenna is transmitting, the remaining antennas act as receivers, leading to a total of *N_t_* = 10 transmitters and *N_r_* = 30 receivers. For breast model imaging scenarios, we enhance reconstruction quality by incorporating measurements at multiple frequencies.

#### 2.1.2. Inverse Solver

Traditional reconstruction methods, such as the BIM or DBIM, estimate the total field solely based on secondary incident fields from the inhomogeneous background. Alternative approaches have been developed to incorporate additional scattering information. The contrast source inversion method (CSI) [28] and the subspace optimization method (SOM) [29,31] utilize both Green’s function operators and scattered field measurements to estimate induced currents within the domain. While CSI processes the induced current as a unified quantity, SOM implements a decomposition strategy that separates it into deterministic and unresolved components. Building upon these concepts, we propose a total field approximation framework as follows:(6)$$E^{tot}=E^{bac}+G_{bD}\cdot C,$$
where ***E****^bac^* denotes the secondary incident field computed with contrast obtained from the image domain at previous iteration or initial guess. The induced current vector ***C*** arises from the contrast difference Δ***χ*** between the contrast of the inhomogeneous background and the true distribution of the image domain ***χ***. This formulation naturally extends the DBIM approach, which can be viewed as a special case where ***C*** = 0. The Green’s function operator of inhomogeneous background is obtained as follows:(7)$$G_{bS}={G_{S}\cdot (I-\chi \cdot G_{D})}^{-1},$$(8)$$G_{bD}={G_{D}\cdot (I-\chi \cdot G_{D})}^{-1}.$$

Real measurement scenarios inevitably involve noise, which affects the induced current distribution. To address this challenge, SOM decomposes the induced current into two components: a primary component containing essential scattering information, and a secondary component predominantly associated with noise effects. This decomposition is achieved through singular value decomposition (SVD) of the inhomogeneous Green’s function:(9)$$G_{bS}=\sum u_{i}\sigma_{i}v_{i}^{*},\;\sigma_{1}\geq \sigma_{2}\;\geq \;\cdots \geq \;\sigma_{N_{r}}\geq \;0.$$

From this SVD-based relationship, we can then compute the induced current as follows:(10)$$C=\sum_{i=1}^{L}\frac{u_{i}^{*}\cdot (E^{sca}-G_{S}\cdot \chi \cdot E^{bac})}{\sigma_{i}}\cdot v_{i}.$$

The integer parameter *L*, ranging from 0 to *N_r_*, determines how many singular values contribute to the reconstruction process. This strategy preserves dominant scattering features while filtering out noise-related components, thereby improving the algorithm’s robustness. Empirical studies suggest that optimal performance occurs when the magnitude of the *L*-th singular value (*σ_L_*) is approximately half that of the largest singular value (*σ_1_*), providing an effective balance between information preservation and noise suppression.

The reconstruction task can be mathematically formulated as a linear inverse problem ***y*** = ***Ax***, where the observation vector ***y*** contains measured scattered field data, ***x*** represents the unknown dielectric property distribution, and ***A*** is the inverse operator. We solve this inverse problem by minimizing a cost function that combines data fidelity and regularization terms:(11)$$F(∆\chi )=∥E^{sca}-G_{S}\cdot \chi \cdot E^{bac}-G_{bS}\cdot ∆\chi \cdot E^{tot}\;{\|}^{2}+{\;\gamma \|∆\chi \|}^{2},$$
where *γ* is the parameter of Tikhonov regularization [34], which is empirically determined.

Solving this optimization problem by S-TwIST, as proposed in our previous work [31], yields the dielectric property changes ∆***χ***, which can be used to update the dielectric properties iteratively.

The tolerance, used as a convergence parameter, plays a crucial role in determining when to terminate the iterative process. The tolerance is set to 10^−4^, as it provided a good balance between convergence speed and reconstruction accuracy in our preliminary tests. Specifically, when the relative update of the solution norm between consecutive iterations falls below 10^−4^, the algorithm is considered converged. If stricter tolerances (e.g., 10^−5^ or 10^−6^) are employed, more iterations would be required, potentially increasing computational cost without yielding significantly improved reconstructions. Conversely, a looser tolerance (e.g., 10^−3^) could lead to early termination and suboptimal image quality.

#### 2.1.3. Adaptive Optimization Method

In the previous literature [31,35], the number of singular values involved in image reconstruction is fixed in the complete reconstructed procedure. The results of this work found that using different numbers of a singular value shows different performances in various noise environments. The results demonstrate that selecting an appropriate number of singular values can achieve a promising noise robustness. This indicates that it is beneficial to have an adaptive method for determining the number of singular values in environments with unknown noise levels. Therefore, an adaptive method for singular value selection was designed. In each iteration, the induced currents and the total electric field for different numbers of singular value are calculated. These total electric fields are then used to reconstruct the domain D in the inverse algorithm. After obtaining all the cost functions, the solution corresponding to the smallest cost function is selected as the starting point for the next iteration. To reduce computational costs, the adaptive method runs at a lower resolution (50% of the original resolution in the x and y axes). Figure 2 illustrates the workflow of the proposed microwave tomography imaging algorithm. During each iteration, the forward solver calculates the electric field data based on initialized or updated dielectric properties. The inverse solver then retrieves the primary component of the induced current using SVD. Based on empirical studies, we select *L* values (L = 5, 10, 15, 20) to balance computational efficiency and reconstruction accuracy. This range represents a trade-off between the instability introduced by excessive *L* values and the inaccurate total field estimation caused by insufficient *L* values. The solution corresponding to the minimal cost function is selected for the next iteration. This process continues until convergence criteria are met. This range represents a balance between the instability introduced by excessive *L* values and the inaccurate total field estimation caused by insufficient *L* values.

### 2.2. Model Parameters

#### 2.2.1. Synthetic Model

In the performance evaluation of synthetic profile reconstruction, the so-called “Austria” profile was used, which includes two solid circles and one annulus, as shown in Figure 3. The target objects are placed within a square area D with side lengths of 2 m. The wavelength of the incident wave is 1 m, and the background medium is free space. The centers of the two solid circles and the annulus are located at (0.3 and 0.6) meters, (−0.3 and 0.6) meters, and (0 and −0.2) meters, respectively. Each circle has a radius of 0.2 m. The annulus has an inner radius of 0.3 m and an outer radius of 0.6 m. The relative permittivity of all objects is set to 2.

The computational domain D is a 2 m × 2 m square and the background is free air. In order to apply FDTD method, the domain D is discretized into 100 × 100 mesh grids. The priori information about the ISP is upper and lower bound of the relative permittivity and the initial guess is obtained from Born approximation. The operating frequency is 400 MHz. The number of transmitters *N_t_* and receivers *N_r_* are 10 and 30, respectively.

#### 2.2.2. Digital Breast Model

The digital breast model represents a more challenging scenario for microwave imaging methods due to its more complex geometric shape compared to the “Austria” profile. This study employs the MRI-derived breast model from the UWCEM Numerical Breast Phantom Repository [32]. In this database, there are nine breast models across four different classes, and all these models are included in the experiment. For each breast model, five 2D coronal samples are selected, ranging from areas near the nipple to areas closer to the chest. Figure 4 illustrates the distributions of relative permittivity and conductivity for this cross-section for one of the 2D digital breast models (breast ID: 070604PA2).

A high-resolution digital breast model with 1 mm resolution is employed for the forward solver to obtain accurate scattered field measurements. In the inverse solver, the model was down-sampled to 2 mm resolution. This coarser resolution serves two purposes: preventing inverse crime (i.e., using identical models in both forward and inverse solvers) and reducing computational complexity during reconstruction.

Within the breast model, conductivity and relative permittivity exhibit a strong correlation. While [36] employs a single-pole Debye model to describe this relationship, we adopt a simpler linear approximation to reduce computational complexity. Through linear regression analysis of the breast tissue dielectric properties, we establish the following relationship between relative permittivity and conductivity:(12)$$\sigma (\varepsilon_{r})=0.0190\varepsilon_{r}-0.0505,$$
where *σ* is the conductivity, *ε_r_* is the relative permittivity. This linear approximation provides computational efficiency while maintaining sufficient accuracy for microwave imaging applications.

The linear model well approximates the relationship between dielectric properties of adipose and fibroglandular tissues as shown in Figure 5. Figure 5 compares the linear relationship in Equation (12) with measurement data, where the scattered points correspond to the 25th, 50th, and 75th percentile values of measured dielectric properties of breast tissue [37,38], similar to the validation approach in [36]. The resulting linear model reduces the number of unknowns in the imaging problem to one parameter, *ε_r_*, per pixel. Reduction to a single unknown parameter at each pixel significantly reduces the complexity of the microwave breast imaging problem. Once *ε_r_* is reconstructed throughout the imaging domain, the dielectric properties at each pixel can be calculated according to the relationship in Equation (12).

Alternative approaches such as multi-pole Debye models could potentially offer higher accuracy but would introduce significant computational overhead without proportional improvements in reconstruction quality. Our focus remains on detecting and characterizing significant tissue contrasts rather than achieving perfect dielectric property reconstruction. The validated linear model’s simplicity contributes to the robustness of our method in noisy environments, which is crucial for clinical applications.

The computational domain D is discretized into 150 × 150 mesh grids. The number of transmitters *N_t_* and receivers *N_r_* are 10 and 30, respectively. The operation frequency is set at 10 different frequency points evenly within the range of 0.8–1.5 GHz. Considering the stability of the optimization, frequency points are arranged from low to high and grouped every three frequency points, with five iterations performed for each group of frequency points. The range of values for the relative permittivity variable is limited from 2 to 55, while the background media is considered to be a non-conductive material with relative permittivity of 10. According to [20], the position of the breast in the image domain is a prior information, thus the algorithm only needs to reconstruct the image within the breast. The dielectric property inside the breast is initialized with the lower bound of relative permittivity.

#### 2.2.3. Performance Evaluation Metrics

To evaluate the quality of the reconstruction algorithms, necessary metrics should be taken into consideration. Error evaluation can be performed by calculating either the relative error of the scattered signals ***E****^sca^*, or the relative error of the reconstructed dielectric properties χ. The relative error of reconstructed dielectric value is defined by the following: (13)$$\xi_{pixel}=\frac{1}{N_{i}}\sqrt{{\left\|\frac{\varepsilon^{rec}-\varepsilon^{ori}}{\varepsilon^{ori}}\right\|}^{2}}\times 100\%,$$
where *ε^rec^* and *ε^ori^* is reconstructed and original relative permittivity, respectively. The factor 1/*N_i_* normalizes the error by the number of pixels in the imaging domain. With the reconstructed dielectric data and total field, the mismatch of observed scattered field data ***E****^sca,true^* and calculated scattered field ***E****^sca^* can be calculated. This error is defined as follows:(14)$$\xi_{signal}=\frac{1}{N_{t}N_{r}N_{f}}\sqrt{{\left\|\frac{E^{sca}-E^{sca,true}}{E^{sca,true}}\right\|}^{2}}\times 100\%,\;$$
where *N_t_*, *N_r_* and *N_f_* are the number of transmitters, receivers, and frequency index.

## 3. Results

In this section, to validate the performance of S-TwIST, several experiments are performed, consisting of Austria profile and breast phantoms. Electromagnetic problems involved in all experiments are simulated numerically using the FDTD 2D model with CPML boundary conditions. The original DBIM algorithm can be considered as a specific case of S-TwIST when *L* = 0.

### 3.1. Synthetic Object Scenario

In the performance evaluation of synthetic profile reconstruction, the “Austria” profile was used. The convergence curves for different *L* values in the 10 dB scenario are shown in Figure 6, with the corresponding reconstructed images in Figure 7. Meanwhile, the convergence curves for the 0 dB scenario are presented in Figure 8, with the corresponding reconstructed images in Figure 9. The results from two different noise levels show that, compared to DBIM, S-TwIST achieves faster convergence when specific L values are used. Notably, the maximum L value (L = 1, 5,…, 30) corresponds to the number of receivers. The convergence curves of the cost function, shown in Figure 6b and Figure 8b, indicate that the adaptive optimization strategy maintains robustness even in high-noise scenarios. Figure 6a and Figure 8a present the relative error curves of reconstructed dielectric properties compared to the true values. Two key observations can be made: First, using a high L value (L = 30) leads to instability in the optimization process, failing to reconstruct the unknown target’s dielectric properties. Second, the proposed adaptive strategy achieves higher reconstruction accuracy in high-noise environments.

Figure 7 and Figure 9 show the reconstructed images under different noise levels using various L values. The results in Figure 7c and Figure 9c demonstrate that using excessive singular values leads to reconstruction failure. Additionally, compared to the well-reconstructed images in the 10 dB SNR environment, images reconstructed in the 0 dB environment exhibit more deformation and distortion. While both the original DBIM and adaptive S-TwIST algorithms show good performance in contour reconstruction, the latter achieves better accuracy in recovering dielectric property values.

The analysis of Figure 6, Figure 7, Figure 8 and Figure 9 confirms that the proposed adaptive S-TwIST microwave tomography imaging algorithm demonstrates superior noise robustness in synthetic object scenarios.

### 3.2. Digital Breast Scenario

Microwave tomography has significant potential in biomedical imaging applications. This section evaluates the proposed algorithm’s performance in breast microwave imaging. Digital breast models present more complex cases than the synthetic patterns discussed in the previous section, featuring non-uniform dielectric properties and intricate details. The diversity of breast tissue composition and density across different samples provides a robust test of algorithm performance. The 45 samples include varying levels of fibroglandular tissue content, from primarily fatty (Type I) to extremely dense (Type IV) breasts. This variation in tissue composition presents different challenges for reconstruction, as higher-density tissues typically result in stronger scattering effects.

The experiment utilized 45 2D breast slices across four categories as imaging targets, with an SNR of 20 dB. Figure 10 presents reconstruction results for three representative samples. The first row shows results from the original DBIM algorithm, the second row displays results from our proposed adaptive algorithm, and the third row contains the ground truth dielectric property distributions. The results demonstrate that while both algorithms can reconstruct the position of breast glandular tissues, the proposed algorithm better preserves the contour information.

Table 1 provides a relative error analysis across all 45 samples. The proposed algorithm achieves promising reconstruction accuracy with a maximum error of 1.8874% and an average error of 0.4847%. The superior performance of the adaptive S-TwIST algorithm can be attributed to its dynamic selection of singular values, which helps maintain stability while preserving fine tissue structure details. The ability to accurately reconstruct both the position and contours of glandular tissues demonstrates the algorithm’s potential for clinical applications where precise tissue delineation is crucial for diagnosis.

To evaluate the performance of the proposed method in more practical scenarios where the breast model is not known a priori, we conducted additional experiments without assuming prior knowledge of the background model. Figure 11 shows the reconstruction results in such a case, where Figure 11a presents the actual permittivity distribution and Figure 11b shows the reconstructed result. The results demonstrate that our method successfully reconstructs the density and location of fibroglandular tissues and adipose tissue even without prior information about the breast boundary. However, due to the lack of prior information about the skin layer, the skin boundary is not well reconstructed.

## 4. Discussion

The convergence curves for different *L* values shown in Figure 6 and Figure 8, along with their corresponding reconstructed images in Figure 7 and Figure 9, demonstrate the effectiveness of our adaptive method. The results confirm that the adaptive approach successfully mitigates noise impact on the S-TwIST algorithm while improving reconstruction accuracy, without requiring prior noise level knowledge.

Two critical scenarios highlight the importance of proper *L*-value selection. First, when L is fixed at maximum in unknown SNR conditions, S-TwIST becomes unstable due to potential noise incorporation. This instability is particularly evident in the 0 dB SNR case, where reconstruction fails completely. Second, selecting too low of an *L* value, while reducing noise impact, introduces excessive bias in total electric field estimation, degrading reconstruction accuracy.

The algorithm’s performance on digital breast phantoms further validates its clinical potential. Across 45 different breast tissue samples, our method achieves superior boundary preservation and detail reconstruction compared to the original DBIM, with average relative errors of 0.4847%. This accuracy holds across various breast tissue densities and compositions, suggesting robust clinical applicability.

From Figure 11b, it can be observed that the fibroglandular and adipose tissue density and location are successfully recovered, demonstrating the algorithm’s robustness even when the background model is not accurately known. However, because the algorithm lacks prior knowledge of the skin boundary, the outer layer (skin) does not appear in the reconstruction. This indicates that although our approach can capture the main interior features under unknown or imperfect background conditions, accurately reconstructing the full breast contour (including the skin) may require additional prior information or constraints.

In clinical scenarios, it is often feasible to obtain approximate skin boundary data via external shape measurements or other complementary imaging modalities (e.g., ultrasound). Incorporating such boundary information into our reconstruction framework could further enhance the fidelity of the outer breast region. Future work will thus focus on integrating these priors to improve boundary recovery while maintaining robust interior tissue reconstruction.

The computational cost of evaluating multiple *L* values presents a limitation. However, this trade-off is justified by the significant improvements in reconstruction quality and noise robustness. The computational overhead of adaptive *L*-value selection has been addressed in our current implementation through strategic mesh resolution management. During the *L*-value selection process, we employ a coarser mesh with a 50% resolution reduction in both x and y directions compared to the inverse solver settings. This optimization significantly reduces the computational burden—the total computational cost for evaluating four different *L* values (*L* = 5, 10, 15, and 20) approximates that of a single iteration with fixed *L* value in the full-resolution reconstruction. This approach maintains reconstruction accuracy while enabling practical implementation of the adaptive strategy.

Looking forward, additional optimization strategies could further enhance computational efficiency. GPU acceleration could be implemented to parallelize the *L*-value evaluation process, particularly beneficial for higher-resolution reconstructions or 3D implementations. Machine learning techniques could potentially be employed to predict optimal *L* values based on signal characteristics, though this would need to be carefully balanced against the robustness of our current adaptive approach. These potential optimizations would build upon our existing efficient implementation to further improve the algorithm’s practicality for clinical applications.

Though this study employs additive white Gaussian noise for performance evaluation, which is a standard practice in electromagnetic imaging research, future work will involve verification against clinical data. It is noteworthy that microwave imaging devices must comply with stringent electromagnetic compatibility (EMC) standards, effectively reducing interference from medical equipment. Furthermore, in many breast imaging systems, a coupling liquid is used to immerse the breast, facilitating electromagnetic wave propagation and providing additional isolation from external noise sources.

Overall, these findings confirm that our method functions robustly under various noise conditions, as evidenced by extensive testing with complex digital phantoms derived from MRI data. This proof-of-concept study serves as a solid foundation for future clinical trials, where experiments with physical phantoms and real patient data can provide further insights into algorithmic refinements. Additionally, while our current work adopts a simplified linear relationship between conductivity and permittivity, future studies will investigate more complex tissue models that more accurately capture the heterogeneous properties of biological tissues.

Following the promising results in 2D simulations, the extension to 3D could provide more comprehensive tissue structure information, as demonstrated in the recent literature [39]. Such an extension would require several key modifications to the current algorithm. For the forward problem, this includes adapting the FDTD solver for 3D geometries, extending Green’s function calculations to three-dimensional space, and implementing 3D mesh generation and refinement strategies. The inverse problem would require modifications to the subspace computation for handling 3D domains and adapting the *L*-value selection strategy to account for increased dimensionality. The antenna array configuration would also need to be modified from the current circular array to a cylindrical or hemispherical array arrangement, with multiple rings of antennas along the vertical axis to capture the full 3D tissue structure. Based on our current 30-antenna circular array implementation, a practical 3D configuration might consist of 3–5 rings with 8–10 antennas per ring to ensure adequate spatial coverage. Our current computational optimization strategies, particularly the adaptive *L*-value selection with reduced mesh resolution, provide a foundation for managing the increased computational demands of 3D implementation. This extension could potentially enhance the algorithm’s capabilities for clinical applications while maintaining its demonstrated effectiveness in noise-robust reconstruction.

## 5. Conclusions

This paper presents an adaptive S-TwIST algorithm for microwave tomography breast imaging. The key contributions include the following: (1) the development of an SVD-based method to extract deterministic contrast sources; (2) the introduction of an adaptive strategy for L-value selection that enhances algorithm stability and reconstruction accuracy in noisy environments; and (3) the comprehensive validation using both synthetic profiles and digital breast phantoms.

The experimental results demonstrate that the adaptive approach effectively addresses the trade-off between noise suppression and reconstruction accuracy. Performance evaluation using digital breast phantoms shows promising results, with average relative errors of 0.4847%, suggesting potential clinical applicability. Notably, the algorithm achieves these improvements without requiring prior knowledge of noise levels, making it suitable for practical implementations.

Future work will focus on experimental validation with physical breast phantoms and optimization of computational efficiency. The successful reconstruction of complex breast tissue structures indicates that this method could contribute to the advancement of microwave-based breast cancer detection systems.

## Figures and Tables

**Figure 1 sensors-25-01429-f001:**
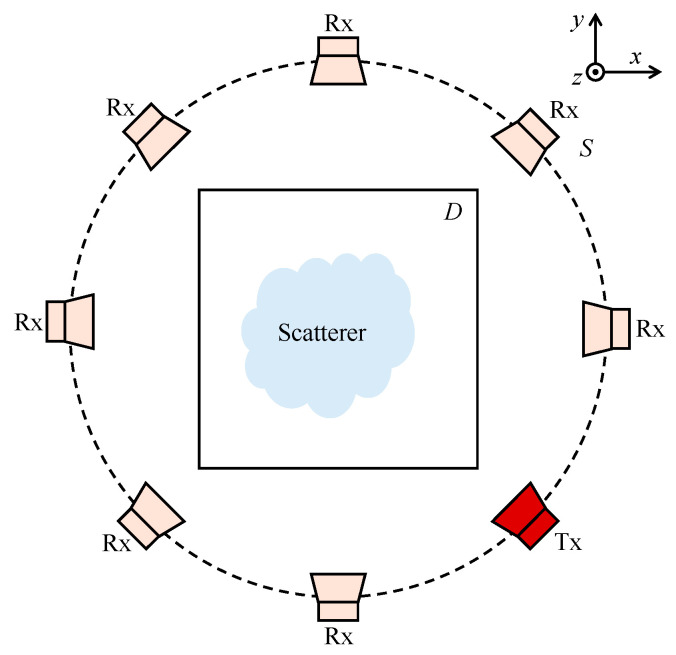
Two-dimensional configuration of the EM scattering problem. The circular sensors array is located outside the domain of interest *D*. Tx and Rx represent the transmitter and receiver.

**Figure 2 sensors-25-01429-f002:**
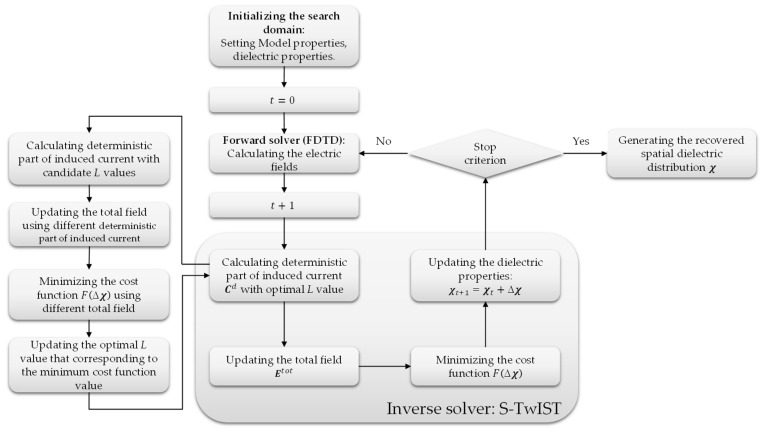
Flowchart of the proposed adaptive S-TwIST algorithm. The algorithm consists of a forward solver for field calculation, an SVD-based inverse solver for induced current retrieval, adaptive L-value selection (L = 5, 10, 15, and 20), and cost function minimization for optimal solution selection.

**Figure 3 sensors-25-01429-f003:**
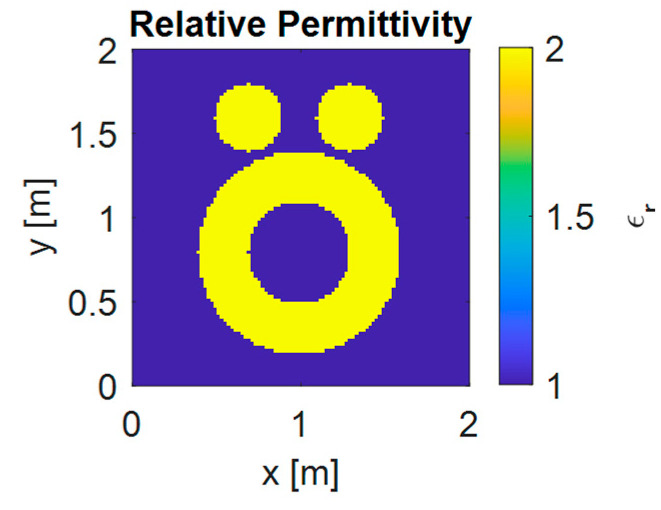
The dielectric distribution of “Austria” profile.

**Figure 4 sensors-25-01429-f004:**
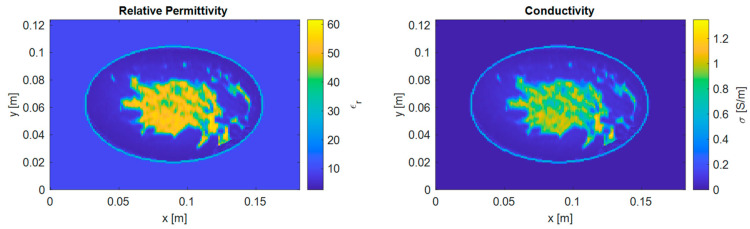
The dielectric properties of the 2D cross-section from the selected digital breast model (breast ID: 070604PA2).

**Figure 5 sensors-25-01429-f005:**
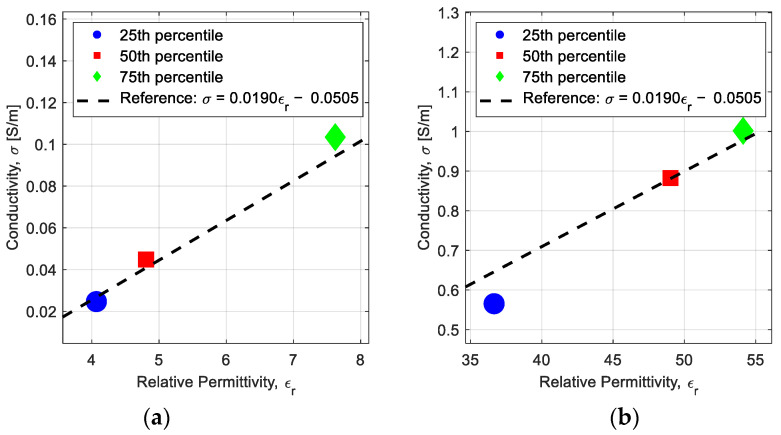
Linear relationship of (2), shown by the black dash line, compared to the dielectric properties that correspond to the 25th, 50th, and 75th percentile values of measured dielectric properties. (**a**) conductivity versus permittivity of adipose tissue. (**b**) Conductivity versus permittivity of fibroglandular tissue.

**Figure 6 sensors-25-01429-f006:**
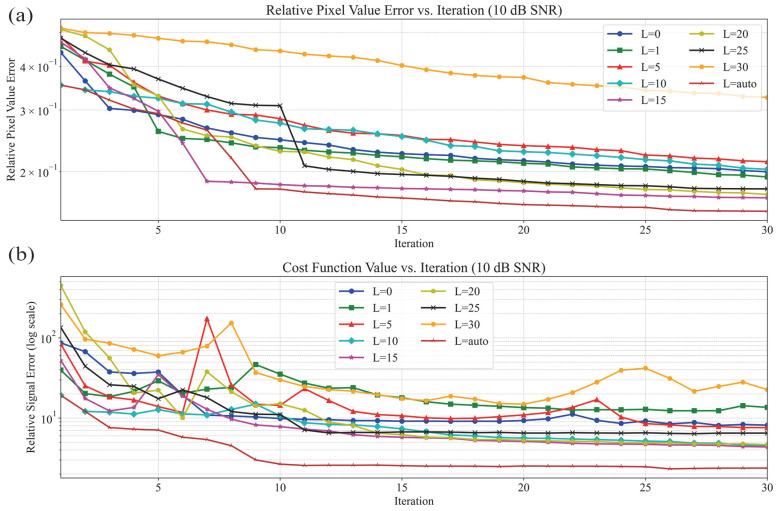
Convergence and relative error curves for different values of L of S-TwIST algorithms when SNR is 10 dB. (**a**) Pixels relative error. (**b**) Base 10 logarithm of cost function values.

**Figure 7 sensors-25-01429-f007:**
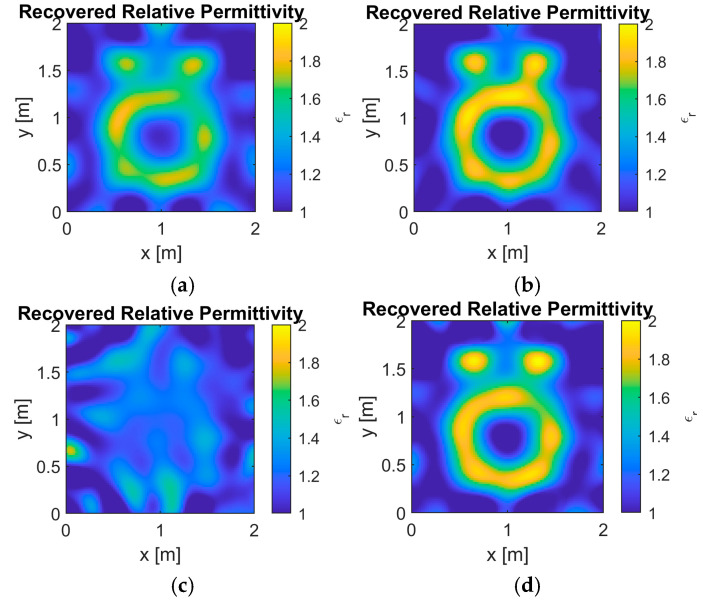
Reconstructed results for Austria profile when SNR is 10 dB. (**a**) L = 0, (**b**) L = 15, (**c**) L = 30, and (**d**) adaptive method.

**Figure 8 sensors-25-01429-f008:**
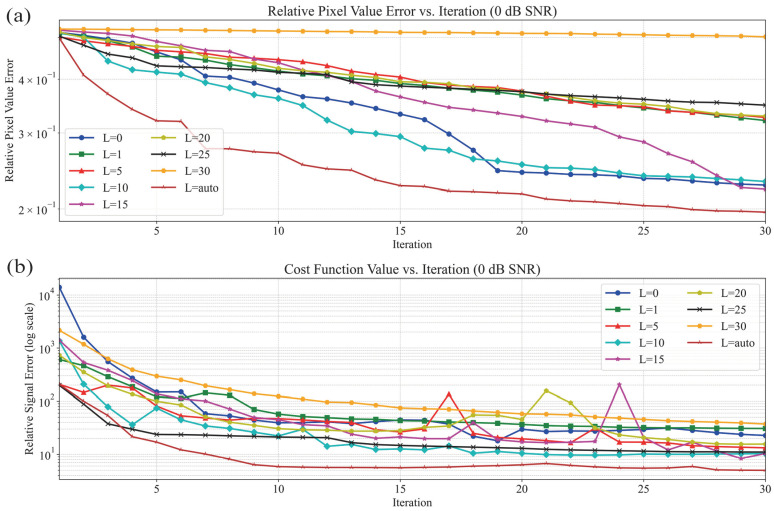
Convergence and relative error curves for different values of L of S-TwIST algorithms when SNR is 0 dB. (**a**) Pixels relative error. (**b**) Base 10 logarithm of cost function values.

**Figure 9 sensors-25-01429-f009:**
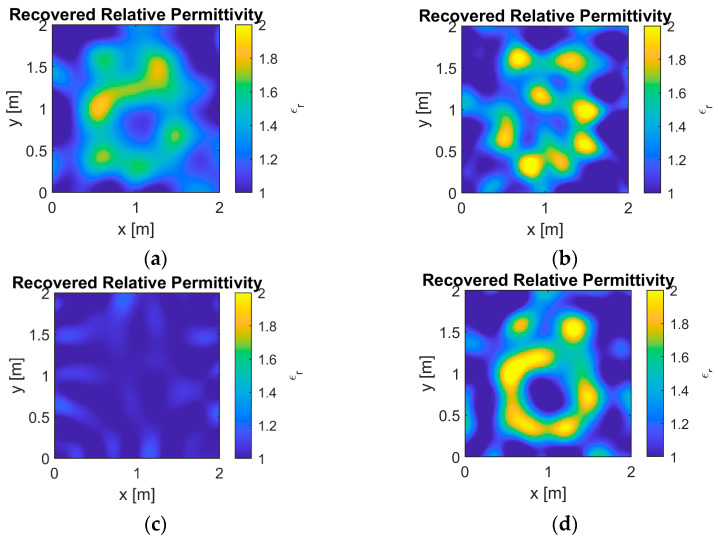
Reconstructed results for Austria profile when SNR is 0 dB. (**a**) L = 0, (**b**) L = 15, (**c**) L = 30, and (**d**) adaptive method.

**Figure 10 sensors-25-01429-f010:**
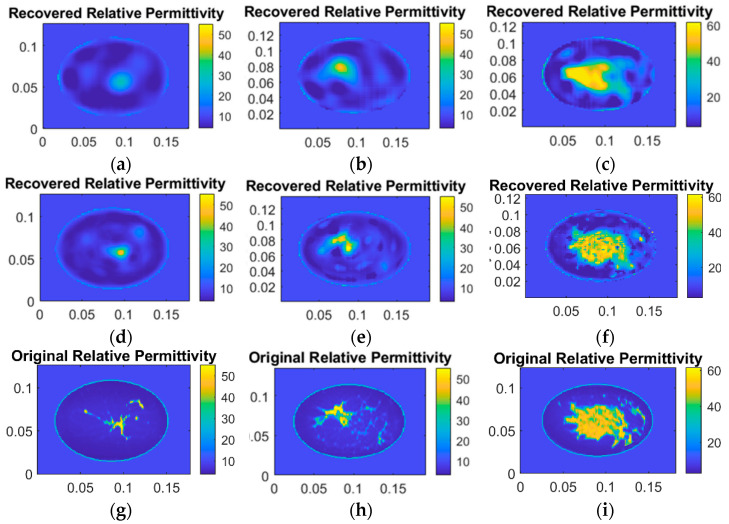
Reconstructed images for three phantoms using two different algorithms when SNR is 20 dB. (**a**–**c**) Reconstructions with original TwIST. (**d**–**f**) Reconstructions with S-TwIST. (**g**–**i**) Original 2-D images of three breast phantoms (mostly fatty, scattered fibro glandular, and heterogeneously dense from left to right).

**Figure 11 sensors-25-01429-f011:**
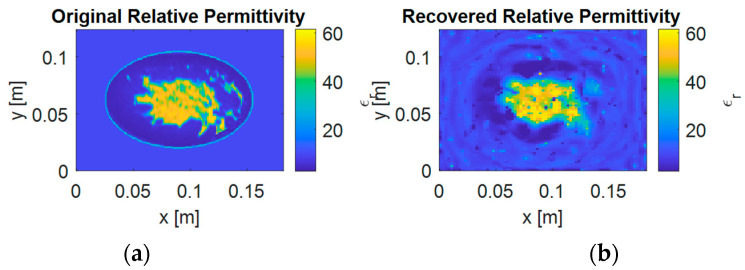
The reconstruction results without prior knowledge of breast boundary: (**a**) the actual permittivity distribution; (**b**) the reconstructed result using adaptive S-TwIST algorithm. While the density and location of fibroglandular tissues are well reconstructed, the skin boundary is not clearly defined due to lack of prior information.

**Table 1 sensors-25-01429-t001:** Breast phantoms error analysis.

Class	Sample	Slice	$\xi_{Signals}$		$\xi_{Pixels}$	
			S-TwIST	TwIST	S-TwIST	TwIST
1	1	1	0.2409	1.04	0.1745	1.0231
		2	0.1652	0.772	0.1366	0.5247
		3	0.2938	0.6672	0.2226	0.4971
		4	0.1904	0.6344	0.3067	0.4145
		5	0.1143	0.5091	0.2242	0.37
	2	1	1.2703	1.4893	0.3903	1.1735
		2	0.0962	1.7478	0.3534	0.4311
		3	0.2019	0.7593	0.6735	0.4835
		4	0.2658	0.7036	0.2091	0.5683
		5	0.1768	0.8414	0.7499	0.4197
2	1	1	0.2047	3.6652	0.2294	2.6978
		2	0.1652	2.1823	0.7834	0.5247
		3	0.3094	0.811	0.3333	0.5355
		4	0.5435	0.7934	0.4897	0.43294
		5	0.2419	1.4264	0.9421	0.3255
	2	1	0.1721	1.604	0.175	1.5034
		2	0.1361	0.8516	0.2089	0.6077
		3	0.3135	0.7275	0.5797	0.5288
		4	0.3124	0.6818	0.3648	0.4168
		5	0.3451	0.7005	0.2942	0.3698
	3	1	0.0702	1.5943	1.0917	3.1522
		2	0.188	2.3823	0.6988	2.6285
		3	3.5646	2.8735	0.6221	2.6954
		4	0.1369	1.89	0.4088	3.2373
		5	0.1417	1.2823	0.3536	2.1186
3	1	1	0.1414	1.5908	0.1451	1.2073
		2	0.1985	0.9111	0.0981	0.984
		3	0.1884	0.8176	0.1602	0.7855
		4	0.1286	0.7541	0.13	0.6428
		5	0.1563	0.69	0.1082	0.6164
	2	1	0.1346	1.7121	0.148	1.4501
		2	0.1652	1.1194	0.1729	1.0131
		3	0.336	1.4661	0.3313	0.6816
		4	0.1771	1.6908	0.358	0.5559
		5	0.19	0.8744	0.3251	0.5349
	3	1	0.0932	0.102	1.8874	1.9887
		2	0.1558	0.2149	0.7364	1.0328
		3	0.3863	0.4783	0.6504	0.7438
		4	0.2217	0.2388	0.5527	0.8352
		5	0.1553	0.2314	0.4998	0.7579
4	1	1	0.1079	0.0965	1.1342	1.0822
		2	0.1333	0.3087	1.0046	1.1498
		3	0.2028	0.278	0.5889	0.6296
		4	0.1507	0.1929	0.7821	0.7149
		5	0.1612	0.2189	0.9822	1.5195
Average error			0.2987	1.0359	0.4847	1.0356
Max error			3.5646	3.6652	1.8874	3.2373

## Data Availability

This manuscript uses numerical breast phantoms from the University of Wisconsin Computational Electromagnetics Laboratory (UWCEM) Phantom Repository. The breast phantoms are publicly available at the UWCEM-Phantom Repository (https://uwcem.ece.wisc.edu/phantomRepository.html) (accessed on 9 March 2024).

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
