# Peer review of "Subspace-Based Two-Step Iterative Shrinkage/Thresholding Algorithm for Microwave Tomography Breast Imaging"

_sensors, 2025, doi:10.3390/s25051429_

Round 1

Reviewer 1 Report

Comments and Suggestions for Authors

This paper presents an interesting enhancement of the conventional subspace-based two-step iterative shrinkage/threshoding (S-TwIST) algorithm. The proposed improvements leverage singular value decomposition (SVD) for contrast source extraction and introduce an adaptive strategy for optimal singular value selection. The results are promising and demonstrate the potential benefits of the proposed approach. However, I have a few concerns that the Authors should address:

  • Page 7: The adaptive choice of L is highlighted as the main novelty of the work. Please provide a more detailed explanation to support and clarify this aspect. Consider including a dedicated scheme or flowchart to illustrate the adaptive selection mechanism effectively.
  • Page 13, Figure 9: What parameters were used for the TwIST algorithm? For the sake of completeness, could you also provide the corresponding conductivity maps?
  • Page 3, Introduction section: To further enrich the state-of-the-art discussion, I suggest considering the following relevant references:
    • Ambrosanio, M., Kosmas, P., & Pascazio, V. (2016, April). An adaptive multi-threshold iterative shrinkage algorithm for microwave imaging applications. In 2016 10th European Conference on Antennas and Propagation (EuCAP) (pp. 1-3). IEEE.

Author Response

Please see the attached file for the response to Reviewer 1.

Reviewer 2 Report

Comments and Suggestions for Authors

The paper proposes an S-TwIST method that aims to improve image reconstruction over traditional methods such as DBIM by using singular value decomposition (SVD).

The algorithm shows very good performance in noisy environments (0-20 dB SNR), with an average relative error of only 0.4847%, without requiring prior information on the noise level. The adaptive strategy for the choice of the L parameter improves the stability and accuracy of the reconstruction under varying noise conditions. 

The paper uses both synthetic and 45 MRI-derived digital sinus models, providing a wide range of test scenarios. However, the proposed paper has some critical issues that need to be resolved. Despite the adaptive strategy, the selection of the optimal number of singular values remains complex, with instabilities in the presence of values that are too high or too low. What future strategies do the authors intend to adopt to reduce the computational complexity associated with the adaptive selection of L, especially for real-time applications?

The tests were conducted on digital phantoms and simulated models, without experimental validation on real clinical data.  

The authors are planning clinical studies to test the algorithm on real data and evaluate its effectiveness in a hospital setting?  

Testing on real data would provide excellent feedback on the implemented model.  The algorithm was primarily tested with additive white Gaussian noise, while in real clinical environments there may be more complex sources of noise.  

How do the authors intend to adapt the algorithm to handle the types of non-Gaussian noise common in medical equipment?  

The approach is based on a 2D model and a linear simplification of the relationship between permittivity and conductivity, which could reduce accuracy in more heterogeneous situations.  

The authors have considered extending the algorithm to 3D models, and what would be the main computational and algorithmic challenges to address?  

The implementation of a three-dimensional model would allow for a more accurate investigation of the processed data.  Of course, I do not expect the authors to implement such a tool.  

I only ask to include a brief paragraph in the text highlighting this possibility, listing the following relevant article in the bibliography: doi: 10.3390/eng5030084.

Furthermore, have the authors considered using more complex models for the relationship between permittivity and conductivity, perhaps based on non-linear or multi-pole approaches?

What alternative approaches, such as machine learning or advanced regularization, could reduce the dependence on initial parameters in more complex reconstructions?

Author Response

Please see the attached file in response to Reviewer 2.

Reviewer 3 Report

Comments and Suggestions for Authors

The manuscript presents an adaptive S-TwIST algorithm in which an adaptive strategy for L value selection has been proposed that enhances algorithm stability and reconstruction accuracy in noisy environments. I think it needs the following revisions before it can be published:

1-Please justify the use of equation (12) and show that it provides a satisfactory relation between conductivity and relative permittivity.

2-Please comment on the use of this technique for 3D imaging of the breast and explain what needs to be modified for practical scenarios.

3-Please show an example similar to the one in Fig. 9 but in which the background model of the breast is not known a priori. Demonstrate how the imaging technique works in such practical scenarios.

Comments on the Quality of English Language

Polishing of the text is required to correct several writing errors. 

Author Response

Please see the attached file in response to Reviewer 3.

Round 2

Reviewer 1 Report

Comments and Suggestions for Authors

The Authors have addressed my concerns.

Author Response

We sincerely appreciate your positive feedback and are glad to know that our revisions have addressed your concerns. Thank you for your helpful suggestions, which have strengthened the manuscript.

Reviewer 2 Report

Comments and Suggestions for Authors

I have no further comments to make. I would like to thank the authors for their cooperation and for addressing the suggestions in depth.

Author Response

Thank you for your time and constructive feedback throughout the review process. We appreciate your acknowledgment of our revisions and are pleased that we could address your suggestions satisfactorily. Your insights have significantly improved the clarity and quality of our work.

Reviewer 3 Report

Comments and Suggestions for Authors

In a part of the response to comment 3, it is written: "it is often feasible to obtain approximate skin boundary data via external shape measurements or low-resolution imaging modalities (e.g., ultra-sound)". Please note that ultra-soud is not a low-resolution modality. Please modify this part.

Author Response

Reviewer #3 Comment:In a part of the response to comment 3, it is written: ‘it is often feasible to obtain approximate skin boundary data via external shape measurements or low-resolution imaging modalities (e.g., ultra-sound)’. Please note that ultrasound is not a low-resolution modality. Please modify this part.

Response: Thank you for pointing this out. We have revised the text accordingly to avoid referring to ultrasound as a “low-resolution” modality. Specifically, at line 480, we now state:

“In clinical scenarios, it is often feasible to obtain approximate skin boundary data via external shape measurements or other complementary imaging techniques (e.g., ultrasound).”

This updated phrasing acknowledges ultrasound’s potential role in skin estimation without categorizing it as a low-resolution modality. We appreciate your feedback, which has helped improve the clarity and accuracy of our manuscript.